# NEURAL NETWORK COST LANDSCAPES AS QUANTUM STATES

## ABSTRACT

Quantum computers promise significant advantages over classical computers for a number of different applications. We show that the complete loss function landscape of a neural network can be represented as the quantum state output by a quantum computer. We demonstrate this explicitly for a binary neural network and, further, show how a quantum computer can train the network by manipulating this state using a well-known algorithm known as quantum amplitude amplification. We further show that with minor adaptation, this method can also represent the meta-loss landscape of a number of neural network architectures simultaneously. We search this meta-loss landscape with the same method to simultaneously train and design a binary neural network.

## 1 INTRODUCTION

Finding a suitable set of weights for a neural network has become one of the most studied problems of modern machine learning. It has presented a significant challenge to computer scientists for whom few successful alternatives to back-propagation are available. It can be difficult to explore very large search spaces efficiently and, worse, optimization may converge to a local minima far from global optimum (Choromanska et al., 2015). Understanding the cost function landscape is also hard, and choosing hyper-parameters and designing neural networks remains mostly a manual process.

As Moore's law approaches its end, two new computing paradigms have been explored, neuromorphic and quantum computers. Quantum computing is based on quantum bits (or qbits) obeying the laws of quantum physics as opposed to the classical bits of today that are based on classical physics. Note that in physics the term classical is used to mean non-quantum and we use this terminology throughout.

Quantum machine learning aims to find an advantage in applying quantum computing to machine learning. Current research into quantum machine learning falls into one of two catgeories. Some quantum algorithms promise a revolution in machine learning in theory, but contain many gaps in their implementation in practice. In contrast, others are more realistic in their method, but struggle to justify a place amongst the well-established methods of machine learning.

In this paper, it is shown that a quantum computer can output a quantum state that represents the entire cost landscape for a given neural network. The method is shown to be versatile and even able to represent a meta-cost landscape of all possible hyperparameters and parameters. Applying it to the connectivities and weights of a binary neural network and simulating the quantum algorithm on a classical computer, we further show that this landscape state can be used for training and meta-training the binary neural network for a small toy problem using quantum amplitude amplification, a standard quantum algorithm.

## 2 RELATED WORK

### 2.1 BINARY NEURAL NETWORKS

Binary Neural Networks (BNNs) are neural networks with weights and activations restricted to taking only binary values, usually $\pm 1$. The greatest advantage of BNNs is in their deployment as using binary provides great advantages in compression and inference time, as well as computational efficiency through the use of bitwise operations. On the other hand they are relatively tricky to train as

the sign function has a derivative of zero nearly everywhere, the search space is discrete, and alternative training methods take significantly longer than non-binarized neural networks. Nonetheless, BNNs have achieved state-of-the-art performance on smaller datasets such as MNIST and CIFAR10 (Courbariaux et al., 2016) but initially suffered when applied to larger datasets such as ImageNet. A popular approach to solving this issue has been to relax the binarisation constraints. This has been achieved by using multiple binary activations (Lin et al., 2017) or by introducing scale factors (Rastegari et al., 2016), both of which result in improvements in accuracy. On the other hand, it has been argued that a better training strategy for BNNs is sufficient to achieve high accuracy on large datasets without compromising on the pure binary nature (Tang et al., 2017). After investigating the accuracy failures of the previous methods, a number of improvements to the BNN training process have been suggested such as changing the activation function, lowering the learning rate and using a different regularization term. These changes helped achieve both high accuracy and high compression rates on ImageNet. Again, this solution is not entirely ideal, as training BNNs is already relatively slow, and a lower learning rate exacerbates this issue. Between the efficient deployment, discrete search space, slow training and relatively small problem size (near-term quantum computers favor problems that require fewer bits), training a binary neural network represents an ideal test case for a quantum computer.

Finally, BNNs have been suggested as a candidate for efficient hybrid architectures through transfer learning. The idea is that a BNN pretrained on ImageNet may be used as a feature extractor for other datasets by retraining a final non-binarised layer. In this way, a hybrid hardware-software architecture can implement the binary part using efficient hardware and the non-binary final layer in software (Leroux et al., 2017).

## 2.2 QUANTUM MACHINE LEARNING

Quantum computers use quantum bits, manipulated with quantum gates in quantum circuits according to quantum algorithms. The advantage of quantum computers over classical computers is that certain quantum algorithms show significantly improved computational complexity compared to the best known classical algorithms. Such improved scaling, combined with the exponentially growing computational power of qubits suggests that (large, error-free) quantum computers would be able to easily handle and process very large amounts of data. Most relevant to this paper is the quantum search algorithm known as Grover's algorithm (Grover, 1996), itself a specific case of another algorithm known as quantum amplitude amplification (Brassard et al., 2002). These algorithms can search for an element of an unstructured dataset of size $N$ in $O(\sqrt{N})$ operations, over the classical $O(N)$. It is important to keep in mind that these are compared to the best-known classical algorithms, and not that they are better than all possible classical algorithms. A recent paper (Tang, 2018) has challenged the presumed superiority of a quantum recommendation algorithm with a new classical algorithm inspired by the quantum method that shows similar scaling. In our case, the optimality of Grover's algorithm has been proven (Zalka, 1999) and so the assumption of its inherent advantage is robust.

Some quantum algorithms are able to efficiently perform $k$-means clustering (Lloyd et al., 2013) and solve linear systems of equations (Harrow et al., 2009), among other such achievements (see Ciliberto et al. (2018) for a review). All of these algorithms require the classical data to be encoded into an accessible quantum form of RAM known as a qRAM. Although there is some work on how this might be done (Giovannetti et al., 2008) it is not known to even be possible to construct a qRAM in an efficient manner for a completely general dataset. To many, this is a significant drawback that cannot be ignored, and places a heavy burden on the feasibility of these methods.

An alternative approach has been to mimic the progress of classical machine learning by using methods classically known to work. Many have taken to using classical computers to train parametrized quantum circuits to perform classification (Stoudenmire & Schwab, 2016a) or to learn generative models (Dallaire-Demers & Killoran, 2018). Some, but not all, of these circuits mimic neural networks in that they are layered and try to utilize non-linearities Killoran et al. (2018). The biggest issue with this approach is the lack of an efficient algorithm for training quantum circuits and so current methods are akin to black box optimization. The motivation is that the output of quantum circuits are known to be impossible to efficiently simulate with classical computers and could therefore provide superior performance on that basis. A slightly different approach to training a perceptron using quantum amplitude amplification has been explored before and its complexity studied compared to classical methods (Kapoor et al., 2016). Previous work has demonstrated and experimentally implemented the use of quantum hardware to perform binary classification, (Neven et al.,

2009) but this is not the same as the method proposed in this paper, as this work is based on a different, more general gate-based form of quantum computation as opposed to the quantum annealing devices of the former.

# 3 QUANTUM COMPUTING

Quantum computing follows the structure of classical computing very closely. Quantum bits, or qubits, are the fundamental unit of quantum information. Their values are manipulated by applying quantum (logic) gates to them in the form of quantum circuits.

Qubits are challenging to manufacture in practice due to the noise-sensitive nature of quantum properties. The biggest such device in existence today contains just 72 highly imperfect qubits, but it is worth noting that progress has advanced at a particularly rapid pace over the past few years and a number are available for public access on the cloud. In addition, simulating the behaviour of qubits using classical computers is difficult, requiring exponentially increasing resources as the number of qubits increases - with an upper limit of 50 (perfect) qubits often cited for the most powerful supercomputers. Therefore, quantum algorithms are almost always defined in terms of their circuit implementation, as opposed to the higher level abstraction of classical algorithms.

## 3.1 QUBITS

Qubits are the unit of quantum information and are fundamentally different to classical bits. Whilst classical bits are completely described as being in one of two states, either 0 or 1, the state of a qubit cannot be fully described by just a single number. It can be in the 0 state, the 1 state or a *quantum superposition* of both. Mathematically the state of a qubit is a two dimensional vector with complex elements and a unit norm. We can write a general form for this vector as $(\alpha \quad \beta e^{i\phi})^T \equiv \alpha |0\rangle + \beta e^{i\phi} |1\rangle$ with $|\alpha|^2 + |\beta|^2 = 1$, $|0\rangle \equiv (1 \quad 0)^T$, $|1\rangle \equiv (0 \quad 1)^T$. Here $\alpha$ and $\beta$ are the probability amplitudes of the zero state $|0\rangle$ and the one state $|1\rangle$ respectively. Qubits cannot be simply read out as classical bits are, but are instead measured. Measurement is a unique feature of quantum mechanics. If the qubit given above is measured, it will be found in the zero state with probability $|\alpha|^2$, outputting a value of 0, and the one state with probability $|\beta|^2$ outputting a value of 1. Therefore measurement of a qubit state always produces a binary outcome, no matter the actual state itself. Measurement is fundamentally indeterministic, probabilistic and irreversible. Upon measurement, the original state is lost along with the values of $\alpha$ and $\beta$ as the qubit collapses to the state $|0\rangle$ or $|1\rangle$ corresponding to the measurement outcome. As a result, the values $\alpha$ and $\beta$ cannot be obtained without repeated measurements of many identical copies of the state. Here $\phi$ is a phase that does not affect measurement outcome, but can be manipulated with quantum gates and play a role in quantum algorithms. Part of the power of quantum computing is the ability to harness superposition to parallelize certain computations and processes.

An important feature of qubits is the way in which they are combined. $N$ qubits are collectively described by a complex vector of unit norm in a similar way as the above, but the length of this vector is given by $2^N$. It is this exponential scaling that makes even modest numbers of qubits unfeasible to simulate on a classical computer.

## 3.2 QUANTUM GATES

In both classical and quantum computing, gates manipulate the states of bits and qubits. As complex vectors, qubit states are transformed into one another by applying complex matrices called operators or simply, quantum gates. This transformation follows the rules of linear algebra and a state $|\psi\rangle$ is transformed into a different state $|\phi\rangle$ by a gate $U$ according to the matrix transformation $|\phi\rangle = U |\psi\rangle$. In order to maintain the stringent requirement of a unit norm, these matrices are restricted to being unitary. A unitary matrix is defined as any square matrix who's inverse is its complex conjugate transpose. Unitarity implies that every quantum gate is reversible, in a manner similar to reversible computing. This fundamental difference in the kinds of operations that can be performed on qubits compared to classical bits is part of the power of quantum computing, but can make analogies to classical computing difficult. Many quantum operations have no classical analogue and conversely, certain simple classical operations (e.g copying the state of a general qubit) are impossible in quantum computing.

COMMON GATES

Just as in classical computing, small sets of quantum gates are universal in that they can be combined to generate any other. It transpires that a small set of quantum gates are sufficient to our work and we choose to list them here, both in terms of their actions and their matrix forms.

The X (NOT) gate flips the state of a qubit from $|1\rangle$ to $|0\rangle$ and vice versa. For qubits in superposition, it swaps the amplitudes of the $|1\rangle$ and $|0\rangle$ states. Its matrix form is

$$X = \begin{pmatrix} 0 & 1 \\ 1 & 0 \end{pmatrix}$$

The Z gate has no classical analogue and takes the matrix form

$$Z = \begin{pmatrix} 1 & 0 \\ 0 & -1 \end{pmatrix}$$

It transforms an arbitrary state $\alpha |0\rangle + \beta |1\rangle$ into the state $\alpha |0\rangle - \beta |1\rangle$. The probability amplitude of the $|1\rangle$ component has changed sign, but the probabilities associated with measurement outcome, as squares of the probability amplitudes, remain unchanged. Note that this still represents a completely different state.

The Hadamard (H) gate also has no classical analogue. It is used to transform qubits from their initial state $|0\rangle$ into the state $\frac{1}{\sqrt{2}} |0\rangle + \frac{1}{\sqrt{2}} |1\rangle$ - an equal quantum superposition of 0 and 1. As a matrix it is

$$H = \frac{1}{\sqrt{2}} \begin{pmatrix} 1 & 1 \\ 1 & -1 \end{pmatrix}$$

The controlled-not (CNOT) gate can be thought of as a generalisation of the classical XOR gate. It performs a NOT gate on a target qubit if a control qubit is in the state $|1\rangle$. We write this as

$$CNOT = \begin{pmatrix} 1 & 0 & 0 & 0 \\ 0 & 1 & 0 & 0 \\ 0 & 0 & 0 & 1 \\ 0 & 0 & 1 & 0 \end{pmatrix}$$

Note that controlled gates can be extended both to arbitrary gates (e.g. CZ) and to arbitrary numbers of control qubits (e.g. CCCNOT).

## 4 METHOD

The main advantage of qubits over classical bits is their ability to be placed and processed in quantum superpositions of states. The key to our method is to use superposition to parallelize the processing of weights in a way not possible classically. Our scheme proceeds as follows:

**General Scheme**
Step 1: The weights are represented in some way by the quantum state of a set of qubits. Setting those qubits into a state that represents an equal superposition of every possible set of weights allows them to define the domain.

Step 2: We then build a quantum circuit analogue of the neural network $U_{QNN}$ that takes in a given set of weights (encoded within qubits as above) and an empty register, and outputs onto the register the corresponding accuracy according to the chosen neural network i.e $U_{QNN}(w, 0) = (w, acc_w)$.

Step 3: Since $U_{QNN}$ is a quantum circuit, inputting weights in superposition form allows them to be processed in parallel. Thus by using the domain-defining qubits as the weights input to $U_{QNN}$ the output will be a superposition correlating all possible weights to their corresponding accuracies. This is what we refer to as the landscape state. We can write this as

$$\frac{1}{\sqrt{W}} \sum_{w \in \mathbb{W}} (w, O_w)$$

where $\mathbb{W}$ is the set of all possible weights, $W$ is its size and $O_w$ the accuracy of the neural network given the set of weights $w$. This is a single quantum state representing the entire landscape of the neural network by correlating every possible set of weights with its resultant accuracy. In the language of quantum physics the weights and the accuracies are entangled.

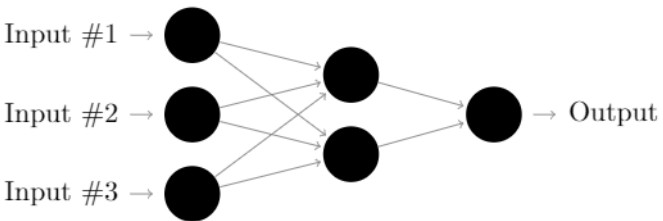

Figure 1: The structure of the BNN we are training. It has a total of eight binary weights.

This method can be adapted in many ways. For example, if just a single weight is set it to superposition and the rest kept to a given value, then the output is the cost landscape of just that one weight conditional on the value of the others. We are not limited to only setting weights in superposition. We note that a meta-neural network with the presence/absence of the connections within the neural network themselves represented by binary parameters can also be created. These meta-parameters can also be encoded in qubits, formed into a quantum circuit and set to superposition. If we set both the weights and the connection meta-parameters to superposition then the output state of the quantum circuit contains an entire meta-cost landscape of every possible weight with every possible connectivity of a neural network simultaneously correlated with the respective accuracy.

### 4.1 EXAMPLE: TRAINING A BINARY NEURAL NETWORK

We demonstrate our method by generating the landscape state for a small binary neural network on simple toy problems and use it to train the network. The advantage of binary neural networks is that each weight can be naturally represented by just one qubit and so are therefore a suitable demonstration given the fundamentally small number of qubits that can be simulated on a non-quantum device.

### 4.2 PROBLEM STATEMENT

We construct two toy problems, both of which are a binary classification on three binary features $x_i \in \{-1, 1\}$ of eight data points corresponding to every $2^3$ arrangement of those features.
In problem 1, the label is given by the function

$$y(x_1, x_2, x_3) = sign(x_3 x_1 + x_2) \tag{1}$$

and for problem 2 the label is given by

$$y(x_1, x_2, x_3) = sign(x_1 + x_2 + x_3) \tag{2}$$

In both cases we define the sign function as:

$$sign(x) = \begin{cases} 1, & \text{if } x \geq 0. \\ -1, & \text{otherwise.} \end{cases}$$

We choose to implement the BNN given in figure 1 meaning that we are aiming to find eight binary weights.

### 4.3 CONSTRUCTING THE QUANTUM BINARY NEURAL NETWORK (QBNN)

To construct a quantum circuit equivalent to the BNN, henceforth known as the Quantum Binary Neural Network (QBNN), every operation in the implementation of a BNN must be mapped to a quantum equivalent. Below we detail each of these and their quantum implementation.

**Representing Numerical Values**

Representing numerical values with qubits is already well established in the literature (Stoudenmire & Schwab, 2016b). Other parts of our construction are, however, incompatible with non-binary input and so we restrict ourselves to the simple case of a binary data input. In this case, the qubit states $|1\rangle$ and $|0\rangle$ represent the values $+1, -1$ respectively. In a quantum circuit, all qubits begin in the $|0\rangle$ state and need only an application of a single NOT gate to be set to $|1\rangle$ where appropriate.

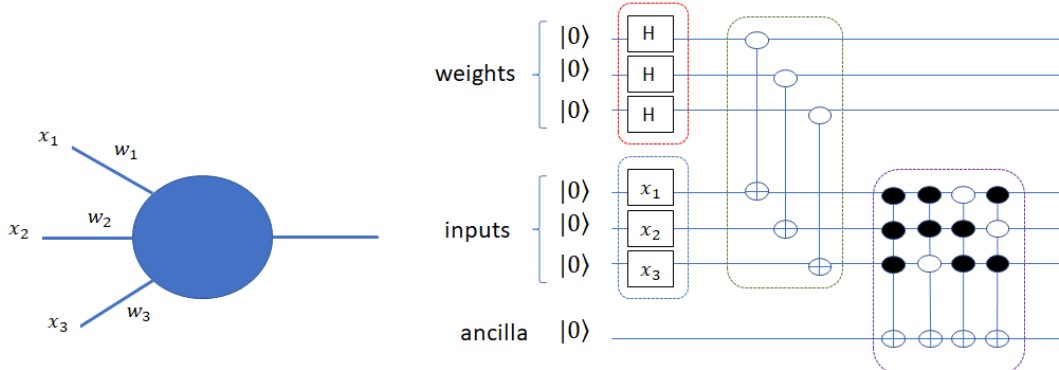

Figure 2: A quantum circuit (right) corresponding to the neuron (left). Quantum circuits are read from left to right just like classical computing circuits. The first group of operations on the weights are initialising them into a superposition of all their values using Hadamard gates (red). The first group of operations on the inputs are encoding the features onto the qubits (blue). They are either blank (if the input values are already -1) or NOT gates if the desired input values are 1. The following three quantum operations are the anti-CNOT gates that weight each input according to its corresponding weight (green). The final set of operations (purple) are a combination of multi-qubit, controlled and anti-controlled NOT gates that together correspond to the sign function. They can be seen to flip the state of the ancilla qubit to +1 if either all the weighted input values are in the state +1, or if any two of them are in the state +1. Ancilla qubits are additional 'helper' qubits.

**Multiplying values by binary weights**
Given two qubits representing binary values $\pm 1$ as described above, we can multiply them using an anti-CNOT gate. An anti-CNOT gate applies a NOT gate to a target qubit if the control qubit is in the state $|0\rangle$ instead of $|1\rangle$. Its truth table is identical to an XNOR gate and outputs $|1\rangle$ if both input values are equal, and $|0\rangle$ otherwise. This truth table matches the truth table of multiplying two binary values and thus performs the same function. It can be constructed using two NOT gates and a CNOT gate. Qubits that encode weights must always be used as control qubits to preserve the values they encode.

**Implementing the Activation Function**
Since the sign function is highly non-linear, it poses the greatest challenge to translate to the linear algebra-based language of quantum mechanics. Generally, the problem can be overcome by the addition of extra helper or 'ancilla' qubits. If we restrict the problem to the special case of binary arguments only, the sign function[1] is reduced to finding whether there exist $N/2$ qubits out of $N$ in state $|1\rangle$. This can be achieved by constructing a quantum analogue of a classical majority function by replacing AND gates with CCNOT gates and constructing OR gates out of CNOT and NOT gates. The number of gates needed scales as the binomial coefficient $N$ *choose* $N/2$. As an example, figure 2 shows a three input neuron and its quantum circuit implementation. Note that this is just a single neuron, and not our entire network. In practice, it works in the same manner as a classical neural network. The activations of each neuron in one layer are then weighted by their own weight qubits and used as input to the next layer and so on. This whole circuit is what we refer to as the QBNN.

**Calculating Accuracy**

For each data point on the training set we must compare the prediction to the label in order to find the accuracy. We initialise a register of qubits to store the predictions. The reversibility of quantum circuits allows us to apply the QBNN for a given data point, store its output value onto its corresponding qubit on the register, perform the same QBNN in reverse order - its inverse - to refresh the other qubits, and continue for the next data point in the training set. This resetting is a common, necessary workaround for small quantum computers and is easily avoided by parallelization given more qubits. For a training set of size $N$, we obtain a register of $N$ qubits containing the predictions of the QBNN for each of them. Since both the labels and the outputs are binary, we can represent the accuracy of each of these predictions by performing a NOT gate on all the qubits corresponding to

---

[1]The sign function with binary arguments is also known simply as the majority function

a data point with a label of 0. Each qubit in this register will then be in the state $|1\rangle$ if it corresponds to a correctly classified data point and $|0\rangle$ if it does not.

### 4.4 QUANTUM TRAINING FROM THE LANDSCAPE STATE

By applying the QBNN over the entire training set with the weights initialized in superposition, our circuit output is the cost landscape state. Training the BNN can be seen as a search for a single state within the cost function landscape, for which we use a quantum algorithm known as quantum amplitude amplification. It is not the first time that quantum amplitude amplification has been suggested as a means to train quantum neural networks (Ricks & Ventura, 2004), but they did not construct the actual details of an implementation such as the method of generating a non-linearity. Quantum amplitude amplification is a technique to amplify the probability amplitudes that correspond to desired state(s) within the superposition and therefore increase the probability of measuring one of these. It works by splitting the space of all states into a 'good' and a 'bad' subspace and rotating their relative probabilities when measured. In this case the 'good' subspace is defined as that which has all the qubits in the prediction register in the state $|1\rangle$ implying that all data points have been correctly classified. It is known that quantum amplitude amplification requires just $O(1/\sqrt{a})$ to search for an entry with an occurrence probability of $a$ (Brassard et al., 2002). Quantum amplitude amplification works by first constructing the amplifying operator, $Q$.

$$Q \equiv -U_{QBNN} \, S_0 \, U_{QBNN}^{-1} \, S_\chi$$

The composite operation, $Q$, is interpreted as a sequence of operations applied from right to left as read in the equation above. $U_{QBNN}$ is our entire QBNN circuit (for all data points), and $U_{QBNN}^{-1}$ is its (matrix) inverse. Since quantum gates are reversible, and every gate we have used is self-inverse, we obtain this by applying all of the gates of $U_{QBNN}$ in reverse order. The operations $S_0$ and $S_\chi$ reverse the sign of the probability amplitudes of the initial state and the target state(s) respectively. In this case, our target states correspond to those with an accuracy of $100\%$ and $S_\chi$ is a controlled-Z gate performed on each of the target qubits. Similarly, the initial state of any quantum computer is defined as having all the qubits in the state $|0\rangle$, and thus we can implement $S_0$ by first applying a NOT gate to each qubit and then applying the same controlled-Z gates as for $S_\chi$.

Figure 3 is a pictorial representation of how quantum amplitude amplification changes the probability distribution of the measured weights. If we write the initial probability of obtaining the correct weights by random as $p$ and the number of successive applications of operator $Q$ to be $k$, it can be shown that the probability of obtaining the optimal weights when measuring the circuit after $k$ amplifications is

$$\sin^2(2k+1)\theta \tag{3}$$

where $p$ and $\theta$ obey the relation $p = \sin^2\theta$ (Brassard et al., 2002). The probability of success is therefore highly periodic in $k$. The problem of training the BNN essentially reduces to a probabilistic search on this one hyper-parameter and its regular periodic landscape. The location of the first maximum, i.e of $k^*$, is inversely proportional to $\theta$ and hence to the probability of obtaining the weights by random. In other words, a harder problem with more weights to search requires a greater number of quantum amplifications to find.

In practical terms the landscape state is a set of 8 weight qubits and 8 prediction qubits. After the search, at the end of the entire process, all the qubits are measured. If the prediction qubits are all in the state $|1\rangle$ the training was a success and the appropriate weights can be simply read off their corresponding qubits.

## 5 RESULTS

We constructed and simulated the QBNN and quantum amplitude amplification circuits on the projectQ framework (Steiger et al., 2018). The use of an actual quantum computer was not possible as the number of gates used during the computation (called circuit depth) exceeds the maximum possible circuit depth for the current generation of imperfect noisy qubits. Furthermore, we use more qubits than are available on current publicly accessible quantum hardware.

For each of the two problems defined, we plotted the probability of obtaining an optimal set of weights against the number of iterations of the quantum amplitude amplification and obtained results, shown in figure 4, that match well with the expected periodic behavior described in equation 3.

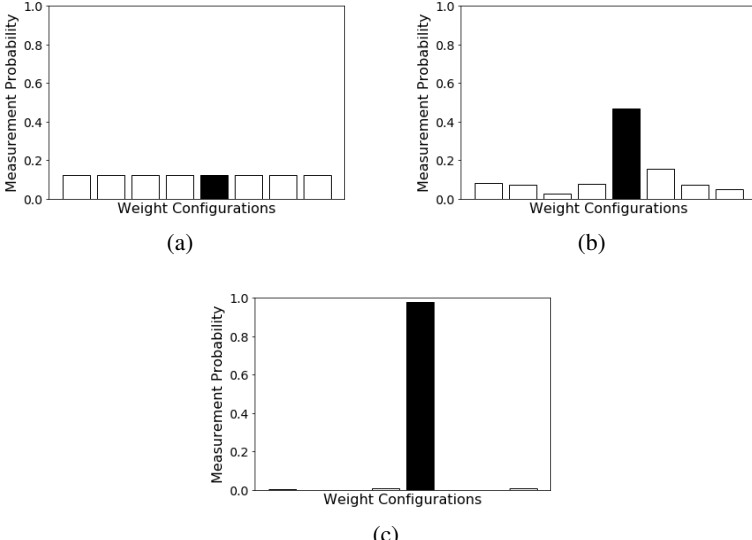

(a)

(b)

(c)

Figure 3: The change in the probability distribution of output weights under quantum amplitude amplification after $k$ iterations where (a) $k = 0$, i.e random , (b) $k$ is sub-optimal (c) $k$ is near optimal. The black data point represents the optimal configuration.

This confirms that a quantum search of the landscape state can indeed be used to train a BNN in exactly the manner as predicted theoretically. We emphasize here that every reference to finding optimal weights means that the BNN has been trained to an accuracy of $100\%$ on the training data.

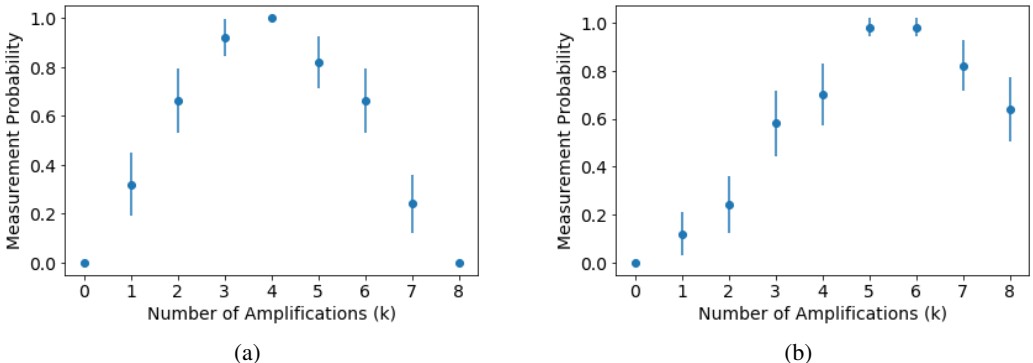

(a)

(b)

Figure 4: A plot showing the relationship between the probability of obtaining an optimal set of weights against the number of quantum amplifications, $k$, for problem 1 (a) and problem 2 (b). Each point represents a simulation of 50 separate runs of the algorithm at the given $k$ and the probability of success of those 50 runs. The error bars represent 95% confidence intervals.

In order to demonstrate the performance of this method in actual training, we follow the simple algorithm described in Brassard et al. (2002) for probing this landscape. This simple algorithm begins with $n = 0$ and chooses a random integer $k$ of quantum amplifications between 0 and $n$. $n$ increases by 1 until the training succeeds. In our experiment, we perform 100 runs of this algorithm and present in figure 5 a cumulative plot of the proportion of these runs that were successful against the number of iterations this algorithm required. We find that training succeeds with a probability over $90\%$ after just 5 steps for the first problem and 6 steps for the second. In order to compare this to a classical search, we search the entire space of $2^8 = 256$ possible sets of weights and find that there are eight and four correct sets of weights (giving $100\%$ accuracy) for the first and second problem respectively. Statistically, if these weights were to be searched through the analogous classical brute

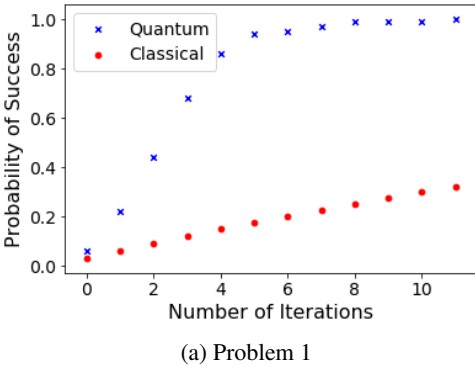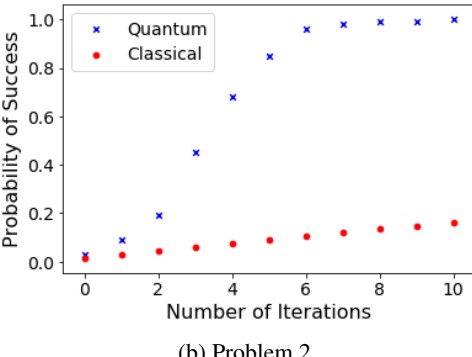

(a) Problem 1           (b) Problem 2

Figure 5: A plot comparing the scaling of a quantum search algorithm over a classical one. The quantum data is the cumulative probability of success over 100 runs of the algorithm. Classical results are analytically derived from the known probability of obtaining a solution by random search. The superior scaling of the quantum algorithm becomes more prominent for harder problems.

force search, one would find that it requires 28 and 57 steps respectively to succeed with a confidence over 90%. This matches our expectation of a quadratic speedup of the quantum search over the classical.

## 5.1 QUANTUM METATRAINING

We then construct a more complex QBNN which can incorporate meta-training by introducing a set of binary indicators that correspond to the presence or absence of a set of connections within the BNN and encode these within qubits in the exact same way as was done with the weights. With the weights and connection parameters both set to superpositions, the output of this circuit is the meta-cost landscape, where weights, connections and accuracy are all entangled with one another. As before quantum amplitude amplification is used to search for the state with all points correctly classified. Again this has been suggested before, but we present a full circuit implementation of this idea (da Silva et al., 2016).

In practice, due to qubit number constraints, we choose to only learn the structure of the first layer of the BNN. The second layer remains fixed. Due to the increased size of the circuit, and the significant increase in computational cost, we did not perform a complete classical search of the space as before but it is clear to see that the space of parameters we are searching has increased and therefore the number of amplifications required has similarly increased. Between 16 and 20 amplifications were found to be sufficient to produce results with a reasonable probability. Figure 6 (a) shows the meta-BNN that was used, and (b) and (c) show two solutions to problems 1 and 2 respectively learnt by our meta-QBNN. It is particularly interesting to note that the learned structures of the two BNN solutions seem to match well with their problem definitions (equation 1 and equation 2). Note that due to our circuit construction a neuron that receives no input will always output $-1$.

## CONCLUSIONS AND FUTURE WORK

We show that quantum superposition can be used to represent many parameters of a neural network at once and efficiently encode entire loss landscapes in a quantum state using just a single run of a quantum circuit. We demonstrate this explicitly for both parameters and hyper-parameters of a BNN, and show that further processing of this state can lead to quantum advantage in training and meta-training. As a training method it possesses significant advantages as it is landscape-independent, has a quadratic speedup over a classical search of the same kind, and would be able to solve statistically neutral problems such as parity problems (Thornton, 1996). It is not, however, without shortcomings.

One potential criticism is the issue of over-fitting. Since our problem is so small, we chose to define a target state as one where the accuracy is 100% on the training set but this is rarely desirable in real machine learning. One solution may be to simply run the quantum algorithm and, upon finding

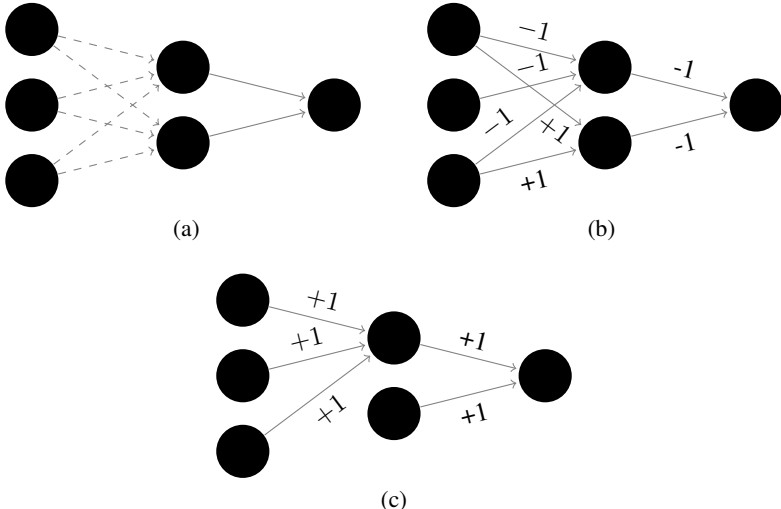

Figure 6: (a) The meta-BNN we train. The dotted lines represent the presence or absence of connections within the first layer. A meta-learned solution to (b) problem 1 and (c) problem 2

a particular set of weights that represents an overfit, run the algorithm again but with a deselection of that particular set of weights. This can be done by simply changing the sign of the probability amplitude corresponding to that state during each iteration of the quantum amplitude amplification. A similar issue is that regular machine learning typically uses batch learning, whilst our method incorporates the entire dataset at once. This too can be fixed by altering our method to use a different batch of the data for each quantum amplitude amplification iteration. This works since no matter what batch we use, a good set of weights should still be amplified by the circuit. In fact, such an implementation is advantageous since it would allow us to use less qubits which in practical terms are limited in number in the near term.

A significant limitation in our method is the requirement that the input is binary, and the poor scaling of the activation function. Both of these problems arise completely from our implementation of the sign function, which could either be improved or replaced entirely with a different binary activation function that could be implemented more efficiently on a quantum computer and would be compatible with non-binary input. There has been progress on creating effective non-linear activation functions by so-called repeat-until-success circuits (Cao et al., 2017). An alternative approach would be to use floating point representations as in classical computing and the quantum equivalent of full-adders, but this would require an overhead in the number of qubits that would take us beyond the limit of classical simulation.

Finally, we note that this method scales poorly compared to backpropagation and that the advantage only appears in like for like comparisons of unstructured classical/quantum searches. The cost function landscape is not unstructured and algorithms such as backpropagation take advantage of this. We conjecture that a quantum search method that applies quantum advantage to structured searches, if it exists, can be applied to the cost landscape in place of quantum amplitude amplification.

Finding ways to harness quantum computers to aid classical machine learning methods in a meaningful way remains an open problem and we present the loss landscape state as a plausible candidate towards this goal. Whilst we used the example of quantum training, the most fruitful approach in the short term is to ask whether some property of the state can be used to glean useful information for classical machine learning methods. This might take the form of understanding the roughness of the landscape, identifying certain features, or even choosing an appropriate learning rate. Further work in investigating the relationship between the landscape as a quantum state and its features from a machine learning perspective would be a step forward in this direction.

ACKNOWLEDGEMENTS

This work was supported by InnovateUK (79852-520140). Concepts and information presented are based on research and are not commercially available. Due to regulatory reasons, the future availability cannot be guaranteed.

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
