# OpenReview forum: "Neural Network Cost Landscapes as Quantum States"
_ICLR.cc/2019/Conference_

### Official Review · AnonReviewer2 · 2018-11-02
**Interesting idea but insufficient clarity and technical depth**

**Rating:** 4
**Confidence:** 4

**Review:**

 This paper proposes a novel idea of outputting a quantum state that represents a complete cost landscape of all parameters for a given binary neural network, by constructing a quantum binary neural network (QBNN). And then the landscape state is utilized to training the neural network by using the standard quantum amplitude amplification method.

Although this idea is interesting, I trend to reject this submission as I think its presentation is unclear and the technical detail is a little difficult to follow. So, the correctness and soundness of this work is difficult to verify. I urge the authors to revise their draft to provide more and clearer technical details.

Detailed comments and questions:

Could the authors further point out that what the scope of the binary features are, {0, 1} or {-1, +1}? To my understanding, it should be {-1, +1}, or the corresponding variables are always +1. In addition, the construction of the “multiplying values by binary weights” module implies that the value should take -1 or +1, rather than 0 or 1. However, at the bottom of page 5, the authors claim that the binary values take +1 and 0. Could the authors clearly explain the term “parameter” and “value”?

Could the author further explain how to construct the majority activation function?
In the part of “calculating accuracy”, the authors mention that “running the QBNN with the weights in superposition for each point in the training set separately” and “there are N qubits containing the prediction of the QBNN”. To my understanding, there are 3 qubits representing the 8 weights, several qubits representing the input values, and N qubits representing the predictions. But how to construct the final landscape state to be optimized with these qubits?

Could the author explain intuitively the main idea of the amplitude amplification method? Specifically, what is the relation between the qubits representing parameters and the qubits presenting prediction results?

During the amplitude amplification process, the probabilities change periodically. How to select the best number of steps k in advanced if we do not known the best parameter? Or how to judge if the training is success?

Overall speaking, I think this paper is interesting. However, the presentation is unclear and I suggest the authors to revise their draft by providing more technical details.

---

> ### Author Response · Authors · 2018-11-25
> **Numerous clarifications + amended paper to include much more technical detail**
>
> We thank the reviewer for recognising the work as novel and interesting and for their constructive comments. We have incorporated these into the new version of the paper. A detailed response to the reviews questions is included below.
> We acknowledge that verifying the soundness and correctness of a paper is challenging if the paper is a little difficult to follow. We strive to create a paper that is very well written and easy to understand for a wide audience. We have altered the paper to include more technical details and improve the presentation as requested by the reviewer. We thank the reviewer for their comments and believe the paper is now easier to read and stronger as a result of this feedback.
>
> The qubits are always either in the state |1> or the state |0>, or in superpositions of the two. What these represent changes depending on context. Most of the time, these represent the numerical values +1/-1 respectively. The only exception is the register of predictions where qubit states |1>,|0> then represent correct and incorrect classifications respectfully. When representing hyper-parameters, the state |1> and |0> represent the existence, or lack of, a connection within the architecture of the NN. We have amended this in the paper for clarity. Otherwise they are simply +1/-1.
> The majority activation function simply uses controlled-gates to count the number of 1’s in a set of inputs to determine if they are in the majority and our quantum circuit does the same. Intuitively the controlled gates are not unlike AND statements and IF statements. E.g. if value x and value y are 1 then value z is 1.  We have given an explicit circuit diagram showing a full quantum neuron explaining every operation and showing what the activation function looks like as a quantum circuit (see figure 2).
> We have made significant amendments to clarify this accuracy calculating issue. The QBNN circuit applies only for a single datapoint, but is reversible as quantum circuits always are. Hence the QBNN is applied for datapoint 1, its valued stored onto one qubit in the register and then the other qubits are reset by applying the inverse circuit which is simply the same circuit in reverse order. We then apply the QBNN for datapoint 2 and store the value in register qubit 2 and so on until all the points have been processed. This process is easily parallelised given a larger quantum computer and is really just a (common) workaround for small qubit numbers.
> Our end result is 8 weight qubits (one for each binary parameter) and 8 prediction qubits (one for each data point). This is the landscape state. The key to this entire process is that these two sets of qubits are quantum entangled. We use quantum amplitude amplification to search the entire landscape state for the subspace where the 8 prediction qubits are all in state 1 (correctly classified) and then measure out the entire set of 16 qubits. If the search was a success, the prediction qubits will indeed all be in the state  |1> and, due to entanglement, the weight qubits will themselves be measured in the state that corresponds to this 100% accuracy.
> In simpler terms, the QBNN is simply a box that takes in an empty register and a set of weights and outputs the accuracy of each data point onto the register and returns the weights. Its quantum nature allows a superposition of all possible set of weights to be a valid input and makes its output a superposition of all possible weights entangled with their accuracies. This is the landscape.
> Searching the landscape for k is an open problem and current approaches use heuristics to give the best results. However, the proposed approach does not need to find the optimal value of k but rather a value that give a high probability of success.

---

### Official Review · AnonReviewer3 · 2018-11-02
**More appropriate for quantum computing literature**

**Rating:** 3
**Confidence:** 4

**Review:**

Review of "Neural Network Cost Landscapes as Quantum States"

Paper summary:

The paper proposes a new algorithm "quantum amplitude amplification"
for training and model selection in binary neural networks. (in which
both weights and activations are restricted to the set -1, 1)

Section 2 references related work and gives some motivation, that some
quantum algorithms scale better (in terms of big-O notation) than
classical algorithms.

Section 3 explains the basics of quantum computing (qubits and quantum
gates).

Section 4 explains the proposed method. There are two toy
problems. The binary neural network has 8 weight parameters. There are
helpful Figures 1-2 which explain the network structure and the
quantum circuit.

Section 5 explains the results of using the proposed method in a
quantum computer simulator (not an actual quantum computer). On the
two toy problems the paper observes quadratic speedups with respect to
a brute force search.

Comments:

A strong point is that the paper is very well-written and easy to
understand.

However there are several weak points which should be addressed before
publication. Major weak points are (1) only (noiseless?) toy data sets
are used, (2) some terms in the paper are unclear/undefined, and (3)
results are unconvincing.

It is not clear that this article should be published in the machine
learning literature. One of the hallmarks of machine learning is a
focus on algorithms for real data sets / problems. In contrast the
focus of this paper is quantum computations on toy data /
problems. Maybe this paper would be better suited for publication in
the quantum computation literature?

The toy problems are explained in section 4.2. Is there any noise or
are these noiseless simulations? How does your model/algo perform as a
function of the noise level? How many data points did you simulate
from the model? (e.g. what is the number of observations in the training set?)

The paper uses the terms "cost landscape" and "meta-cost landscape"
without explicitly defining them. Equations should be added to clarify
these terms.

Results could be made more convincing by
1. using a real quantum computer.
2. using real data rather than toy data.
3. adding error bars or confidence intervals to Figures 4-5.
4. using a more appropriate baseline -- why not try the algorithms mentioned in section 2.1?

Figure 3 could be clarified by providing ticks and labels on the x
axes.

---

> ### Author Response · Authors · 2018-11-25
> **Defended the suitability of a disruptive but still emerging field as quantum machine learning in an ML journal + amended paper where appropriate `1/2**
>
> We thank the reviewer for the constructive feedback and for recognizing the work as new and the paper as very well written and easy to understand. We have altered the paper to address all points raised by the reviewer.
> We thank the reviewer for the opportunity to clarify points in the paper that can be improved to make the manuscript stronger.
> 1) Noiseless, synthetic datasets, real data/problems as a hallmark of ML. As of today, the available quantum computers are small in size. It is this small size that restricts the magnitude and characteristics of the data that we can use for training/testing. For example, it is not physically possible to fit all the data in a single PASCAL VOC Dataset image onto a current quantum computer. However, larger and larger quantum computers are being commercially built by companies including Google, IBM, Intel and Rigetti. This restriction on size will disappear in the near future.
> We also argue that synthetic data, noiseless or simplified datasets have played and continue to play a crucial role in the advancement of the field of machine learning and we expect a similar path to be followed for quantum machine learning. Datasets such as Tic-Tac-Toe Endgame Dataset, Connect-4 Dataset, Iris, NMIST and more recently Atari 2600 games have enable theoretical and experimental advancement in machine learning that reflects the state of currently available hardware. In this paper we have chosen the largest datasets, given the current restrictions on quantum hardware, in order to open up and explore a new research area. This is an entirely new paradigm that is developing extraordinarily rapidly. Such a field requires the foundations to be set through proofs of concept such as these. Classical machine learning followed the same trend before the widespread availability of computing power; this can be reflected in the conference call for papers which includes theoretical issues in learning.
> Due to the limited size of the problem we investigate, we do not add noise. However, it is important to consider what noisy data would achieve. We stress that the quantum neural network simply mimics its classical counterpart with greater parallelisation. If we are to input noisy data the accuracy that is output would be the exact same as for a classical neural network with the same noisy data, activation function and weights. The method for generating the landscape state will not change and the quantum search will still have a quadratic speedup over its classical equivalent.
> 2) some terms in the paper are unclear/undefined. The paper has been updated and all terms are clearly defined with additional explanations added
> 3.1) Real quantum computer. At the time of writing several companies are developing commercial quantum computers with a variety of qubits counts. Rigetti 128, Google 72, IBM 50 and Intel 49, unfortunately these machines are not available to the public. The largest publically available quantum computers are IBM’s 20 qubit and Rigetti’s 19 qubit machines. These would not be large enough to run our proposed approach however, we expect to be able to do this in the near future. We agree that this will make the paper stronger. We perform our experiments in quantum simulations to enable us to explore the theoretical and experimental advancement of this research topic, this is standard practice.
> 3.2) Real data. This has been addressed above
> 3.3) error bars. These have been added
> 3.4) baseline. What we have done is taken a classical neural network and created a quantum analogue and taken a classical search method and used a quantum equivalent. We believe that the most relevant and direct baseline comparison here is the original classical method to our quantum equivalent.  As for the algorithms mentioned in section 2 – our method is underpinned entirely by an adapted version of one of them – Grover’s algorithm. Many of the others rely on as-yet unfounded assumptions on the data itself, primarily that it will be stored and accessible as quantum amplitudes. Our primary motivation was to investigate realistic near-term methods that do not rely on these assumptions.

---

> > ### Author Response · Authors · 2018-11-25
> > **Response to reviewer 2/2**
> >
> > The reviewer asks the question should this paper be published in the machine learning or quantum computation literature.
> > We believe that it is very timely for quantum machine learning papers to be published in high impact machine learning conferences for 3 reasons; 1) growing interest from the machine learning community, 2) maturity of quantum hardware and 3) potential impact.
> > 1) Quantum Machine Learning is of increasing interest to the wider machine learning community. This is reflected by recent quantum machine learning workshops at NIPS and KDD, the increased publication of quantum related papers at high impact ML conferences and new collaborations between the machine learning and quantum communities. These collaborations include the MIT’s quantum learning initiative with IBM focused on enhancing AI with quantum, Alibaba’s Cloud AI collaboration with Chinese Academy of Sciences Innovative Center in Quantum Information, Googles Quantum AI group’s collaborations with University College London/MIT, and Google/Deepmind working with the University of Basel on quantum chemistry.
> > 2) The maturity of quantum hardware.  Quantum computers have moved from the research lab into commercial companies with IBM, Google, Intel, Microsoft, D-wave, Rigetti, IonQ, Nokia Bell Labs, Oxford Quantum Circuits and many more manufacturing small scale devices that can run quantum algorithms. These near term devices are enabling algorithmic experimentation and have led to the development of new algorithms. These are key milestones on the road to larger devices that can outperform classical computers.
> > 3) Potential impact. Machine learning algorithms are fundamentally shaped by the hardware they run on, this has been the case for single CPU, multi core CPU, GPU and TPU. The most notable is the disruption that GPU’s have had on machine learning and the way they enabled deep learning algorithms. As the end of Moore’s law approaches, hardware manufactures are exploring post Moore’s law computing paradigms. One of the most promising paradigms is quantum computing as it offers polynomial or even exponential speed ups over classical computing for some problems. We believe that quantum processing units have the potential to disrupt machine learning in the same way that GPUs disrupted the research area. This impact can only be realized through engagement with the machine learning community.

---

### Official Review · AnonReviewer1 · 2018-11-03
**Review TLDR: good paper interesting subject good fit for ICLR maybe even better for QIP**

**Rating:** 5
**Confidence:** 5

**Review:**

Neural Network Cost Landscapes as Quantum States

The authors describe a method where a deep learning framework can be quantised, this is done by considering the two state form of a Bloch sphere/qubit and mapping binary neural network. onto the quantum object creating a quantum binary neural network

I have to say I liked the paper, it is indeed novel and I haven’t seen this anywhere else. More then that, it addresses the quantum aspects of deep learning which has only recently started getting so much attention rather the regular machine learning algorithms. And such I think it’s a good fit for ICLR.

With that I have a few concerns/ nitpicking issues I would like the authors to address if possible
1. While authors show basically a discreet network (much like Soudry’s work) there has also been recently a show of continues variable networks (https://arxiv.org/pdf/1806.06871.pdf), how does your scaling compare to that ? Could one think of this is the continuum limit (albeit you showed a *very* small system) of your model ?
2. As a general style remark there is a lot of introduction on quantum information and I wondering if authors could just reference classical text such as Isaac Chuang, Michael Nielsen or for the CS flavour Classical and Quantum Computation  ? I would have liked to see all 3.1,3.2 maybe in an appendix and a lot more details on the experiments and setup for example
3. At the end of section 3.2 authors mention the lack of correspondence principle in some quantum systems, I would be happy for a refinement in the aspect of quantum computing that is true and also in general quantum mechanics but there is a huge body of work and an entire field dedicated to just that, settling the difference in correspondence principle and it’s meaning (for example Chaos and the semiclassical limit of quantum mechanics (is the moon there when somebody looks?) by Michael Berry. While this is defiantly far from a deal breaker I would be happy for a bit more clarification on subtle difference. My guess is that authors are referring to the fact that an essential part of quantum computing is the lack of correspondence that can also be taken advantage of for quantum parallelism/ quantum speed ups ? Same for the last paragraph of 3.1.
4. In the method section  all of this procedure is described for a fault tolerant machine, can you say something about coupling to error correction codes for near term quantum devices ?
5. On the same note, there is a feeling that non linearities are swept under the rug, do you have a hunch how can one use non linear activation functions ? (General question, you don’t really have to answer and can take the fifth )
6. In the problem statement (4.2)  x is in fact the binary representation of the qubit state ?
7. In your method Fig. 2, should there be some sort of measurements ?
8. Can you elaborate what exactly goes into the U gates and how are they constructed in Fig.2 ? Or is it some  oracle model that can do all the actions
9. You report a quadratic speedup, is there some relationship to random walk  is this in fact a case of a RW search over the parameter space ?

---

> ### Author Response · Authors · 2018-11-25
> **Addressed specific comments and updated paper with numerous clarifications**
>
> Thank you for recognizing the paper as novel and a good fit for ICLR. As the reviewer points out, quantum aspects of deep learning is a new area of research that is gaining attention and we hope this nascent area can grow into an impactful research topic by reaching the wider machine learning community through publication at high quality machine learning conferences. It is clear that quantum computers have the potential to be disruptive to machine learning and we believe this is a timely opportunity to bridge the gap between the two research communities. Again we thank the reviewers for recognizing that it would be of interest to the ICLR community.
>
> We preface this response with a note that all the clarifications and requested additional technical details have been included in the new draft.
>
> 1.	A reference to the work has been added. This paper is very interesting as, like our work, the formulation of the algorithm is influenced by the underlying hardware. Our formalism is based on qubits and theirs is based on continuous variables and quantum optics which has a different way of processing the information. As such as direct comparison is not straight forward. A comparison is made even harder by the fact that the method described in the paper does not use a quantum training scheme as we do. The training is actually done via a classical optimization of some parametrised circuit from which a source of quantum advantage is unclear. It is not clear how we might compare this with our fully quantum search.  The scaling of their optimization depends almost entirely on the landscape itself whereas our method is completely landscape independent and is a function only of the proportion of the sizes of the ‘good’ and ‘bad’ subspaces – i.e the proportion of weights that would give a good classification compared to those that would not. It is unfortunately a case of comparing apples with oranges.
>
> 2.	We have modified the paper to reflect this comment and added significant detail to the set up and technical detail. The aim of this is to provide sufficient detail and depth for readers with an existing knowledge of quantum physics. Striking the balance between referencing background and including background information is challenging. We are acutely aware that quantum mechanics is a complex topic and experts in the machine learning community may not have a physics background. To make the paper as accessible and self-contained as possible we have resolved to keep an introductory section.
>
> 3.	Yes, the reviewer’s assumptions are correct. Without superposition and entanglement this method does not work and there would not be any quantum advantage. The issue with semiclassical descriptions is that by definition they are quantum systems that are amenable to a classical description. By their very definition that means that they are not suitable for quantum computation. For this reason they are generally not considered by the quantum computing community.
>
> 4.	Our method is fully compatible with quantum error correction schemes. Unfortunately, to simulate them would require a large qubit and gate overhead which is not computationally tractable with today’s classical computers. As larger quantum computers become available we will be able to experimentally evaluate the algorithm.
>
> 5.	We have changed the paper to make clearer references to the issues around non-linearity. We address non linearities in the paper by including extra ancilla qubits and taking a subspace of them (i.e ignoring some qubits). This can be seen in the new figure 2. The activation function takes 4 qubits (3 input + 1 ancilla) and only one “comes out”. Alternative directions of research on this topic are discussed in the conclusions. This includes repeat-until-success circuits but these are not deterministic and seem to require very large numbers of repetitions for each non-linearity.
>
> 6.	The x’s simply represent binary values +/- 1. This has been clarified in the new draft.
>
> 7.	Measurement does not talk place until the end, after the quantum amplitude amplification has taken place. The paper has been modified to make this technical detail clearer. As in a classical neural network the output of that quantum neuron would be fed forward (without measuring) as input to the next layer. Measurement would destroy the entanglement being built so is not performed until the end when the search has been completed.
>
> 8.	There is not an oracle, they are gates. The paper's text and figure 2 have been amended to give the explicit forms of every process in a single quantum neuron.
>
> 9.	Yes this is correct in the sense that all quantum algorithms can be described as a quantum walk.

---

### Meta-Review · Area_Chair1 · 2018-12-11
**ICLR 2019 decision**

**Confidence:** 4
**Recommendation:** Reject

**Metareview:**

This paper studies the problem of training binary neural networks using quantum amplitude amplification method. Reviewers agree that the problem considered is novel and interesting. However the consensus is that there are only few experiments in the current paper and the paper needs more experiments on different datasets with comparisons to proper baselines. Reviewers opined that the paper was not so easy to follow initially,  though later revisions may have somewhat alleviated this problem.